# Mapping the Spatial Distribution of the Rumen Fluke *Calicophoron daubneyi* in a Mediterranean Area

**DOI:** 10.3390/pathogens10091122

**Published:** 2021-09-02

**Authors:** Antonio Bosco, Martina Nocerino, Mirella Santaniello, Maria Paola Maurelli, Giuseppe Cringoli, Laura Rinaldi

**Affiliations:** Department of Veterinary Medicine and Animal Production, University of Naples Federico II, CREMOPAR Campania Region, 80137 Naples, Italy; nocerinomartina@gmail.com (M.N.); mirella.santaniello@libero.it (M.S.); mariapaola.maurelli@unina.it (M.P.M.); cringoli@unina.it (G.C.); lrinaldi@unina.it (L.R.)

**Keywords:** *Calicophoron daubneyi*, ruminant, spatial pattern, prediction, southern Italy

## Abstract

Rumen flukes (*Calicophoron daubneyi*) represent a growing threat to the animal health, productivity and welfare of ruminants. The present study aimed to assess the spatial distribution of *C. daubneyi* infections in ruminants and to develop a predictive model of the environmental suitability for rumen flukes in a Mediterranean area. A cross-sectional coprological survey was conducted in 682 sheep, 73 goat and 307 cattle farms located in the Basilicata region (southern Italy). Faecal samples collected were analysed using the FLOTAC technique. Geographical Information Systems (GIS) and statistical models were developed to determine environmental risk factors and to delimitate the areas at highest risk of infections in small ruminants. The results showed 7.9% (95% CI 6.05–10.27) of sheep farms, 2.7% (95% CI 0.48–10.44) of goat farms and 55.0% (95% CI 49.62–60.99) of cattle farms were infected by *C. daubneyi*. The areas with high predicted risk were situated in the western part of the region. The soil texture, land use and the presence of streams and brooks were the variables statistically significant (*p* < 0.05) in explaining the *C. daubneyi* distribution in the study area. The study confirms the importance of geospatial technology in supporting parasite control strategies in livestock and demonstrates that a combined use of different geostatistical techniques can improve the prediction of the *C. daubneyi* infection risk in ruminants.

## 1. Introduction

*Calicophoron daubneyi* (rumen fluke or paramphistomes), the causative agent of paramphistomosis, is a ubiquitous Trematoda that resides in the digestive tract of ruminants. This pathogen has traditionally been considered of low clinical significance, causing a subclinical pathology, when livestock animals are maintained in their best nutritional and health status, as is usual in Europe [1], e.g. France [2,3] and the UK [4,5,6], thus representing a potential impact on ruminant health, productivity and welfare. Heavy parasite burdens commonly compromise livestock production through reduced feed conversion efficiency, weight loss and decreased milk yield, thus incurring economic losses with elevated morbidity and mortality [2,5].

Increased prevalence of this infection has been reported in various European countries [3,7,8,9,10]. Ruminants become infected by the ingestion of encysted metacercariae on pasture. The higher rate of prevalence may reflect an increased risk of infection, possibly triggered by changes in climatic conditions (i.e., increased temperatures and rainfalls) favoring higher transmission rates of these parasites [11]. The life cycle of *C. daubneyi* needs an intermediate snail host (e.g. *Galba truncatula*) to be completed [12] which is also the predominant intermediate host of the liver fluke (*Fasciola hepatica*). For this reason, the transmission of *C. daubneyi* is associated with the presence of freshwater gastropods, in which the parasites multiply. The amplifying efficiency of the intra-molluscan cycle determines the number of metacercariae produced and the associated risk of infection in grazing animals [1].

Recent studies on the spatial distribution of *C. daubneyi* explain the geographical heterogeneity in the probability of livestock infection due to environmental factors influencing the parasite’s life cycle [7,8,13]. However, as environmental conditions in the Mediterranean area are very varied, such as the orographic/hydrographic characteristics and the coverage of soil, the risk of infection also varies. For this reason, the environmental parameters must be considered in epidemiological studies. Therefore, geospatial health represents the ideal approach for infection risk detection and consequent spatial distribution monitoring activities through regular parasitological diagnosis. Compared with *F. hepatica*, there is limited scientific knowledge about the spatial distribution of *C. daubneyi* especially in the Mediterranean area. For these reasons, this study aimed to analyse the spatial distribution of *C. daubneyi* infection in cattle, sheep and goats in southern Italy in order to identify the main environmental conditions favorable for the development of rumen flukes and their intermediate hosts. The results also allowed us to develop a predictive model of the geographical distribution of rumen flukes in small ruminants in the study area.

## 2. Results

The results showed 7.9% (95% CI 6.05–10.27) of sheep farms, 2.7% (95% CI 0.48–10.44) of goat farms and 55.0% (95% CI 49.62–60.99) of cattle farms infected by *C. daubneyi*. Regarding the *C. daubneyi* eggs per gram (EPG) of faeces, the mean value was 109.1 (minimum = 10.0; maximum = 1405.0) in cattle farms, 76.0 (min = 2.5; max = 565.0) in sheep farms and 45.0 (min = 30.0; max = 60.0) in goat farms (Figure 1).

The density geographical distribution of rumen fluke infection in cattle, sheep and goats was calculated using Kernel density analysis and mapped in Figure 2a; it evidenced two pronounced spatial patterns at north and at south-west of the study area.

The Hot Spot analysis of goat and sheep farms revealed that 69 farms represented statistically significant hot spots at the 99% upper confidence level (CL_U_), 17 at the 95% CL_U_ and 17 at 90 CL_U_. None of the farms represented statistically significant cold spots at the 99% CL_U_. Instead, 1 and 38 were significant at the 95% CL_U_ and 90% CL_U_, respectively (Figure 2b).

The spatial distribution of *C. daubneyi* generated by the RF algorithm is presented in Figure 3. The map shows the predicted presence/absence of the parasite in sheep and goats for each 100 × 100 m cell of the raster. In the RF model implemented, infection by *C. daubneyi* can be identified using predictors of hydrology, topography, land use and soil texture. A table showing the importance of variables detected to be significant was generated by the algorithm (Table 1). In addition, predicted values were compared with the observed values of positivity by the Forest-based model, and the accuracy in prediction was 80%.

Overall, the findings of the Kernel density, the Hot Spot analysis and the RF spatial predictive model identified areas with high predicted risk in the western part of the region.

Regarding the statistical analysis, the Spearman’s (*r_s_*) test results were in agreement with the ANOVA test results and both showed that areas with loamy soil texture located at an elevation between 500 and 1500 masl with a slope of less than 15° were positively correlated with the presence of *C. daubneyi*. In these areas emerged an extended presence of broad-leaved forest, transitional woodland-shrub and pastures. The Spearman’s rho correlation coefficients and the *F*-statistic values from the one-way ANOVA test are listed in Table 2.

Only three variables positively correlated with the presence of *C. daubneyi* were included in the best-performing model generated by the logistic regression analysis: the average elevation, the presence of streams and brooks, and the clayey over fragmental soil texture (Table 3).

## 3. Discussion

The analysis of the spatial distribution of *C. daubneyi* conducted in the Basilicata region (southern Italy) showed that the prevalence of rumen fluke infection was lower in sheep and goats (7.9% and 2.7%, respectively) than in cattle farms (55.0%), in accordance with previous findings by Jones et al. [14] and Naranjo-Lucena et al. [15]. Low prevalence and EPG values in sheep farms were also observed in previous studies conducted in different areas of southern Italy by Cringoli et al. [7] and Musella et al. [8], with a prevalence of 16.2% (EPG mean = 52.0) and 14.0% (EPG mean = 3.5), respectively. The fluctuations in prevalence of infection among the different ruminant species could depend on different factors related to animal behaviour and farm management. The rumen fluke prevalence in goats was lower than in other ruminants, probably due to the fact that goats tend not to drink from drying water bodies where the levels of metacercariae tend to increase [16]. Furthermore, the difference in prevalence between sheep and cattle may be due also to some differences in the farm management such as the intense use of flukicide drugs and small size of grazing areas in ovine farms compared to cattle farms. Other authors also suggest that different factors such as production system, breed, animal density and age group might affect the prevalence and intensity of *C. daubneyi* infection in cattle, although the statistical analysis frequently did not show these variables as significantly associated with rumen fluke infection [9,17].

In the present study, Hot Spot mapping was used as a basic output of infection prediction. It used data on all farms’ infection status to predict where spatial patterns of parasitosis may occur in the future; nevertheless, it does not provide information about factors that are considered to cause the phenomenon. However, this technique is consistently included among the best for predicting spatial patterns of infection (especially in large-scale study cases where only parasite presence/absence data are available). Conversely, the RF technique provided a more likely and detailed prediction since the algorithm processed different forms of explanatory variables for the Forest model construction (Figure 4); however, this methodology also showed some limitations. Firstly, information about the feature of each explanatory variable more correlated to the presence of *C. daubneyi* was not provided. Although the variable importance table generated by the RF algorithm was useful to understand which variables were driving the results [18], the method of determining variable importance was biased in favor of the variables with more levels, as reported by Strickland et al. [19]. In addition, the infection prediction map returned as output, showed some “holes” in the locations at which the categories in the prediction dataset did not exist in the training dataset.

The model generated by the logistic regression instead not only considers the significant environmental variables derived from the univariate analysis, but also determines the presence and strength of any relationships between the infection status (positive/negative) of the farm and each feature of environmental variable. This last predictive technique differs from the previous two since it uses a not-punctual approach for the extraction of the variables. More specifically, for sheep and goat farms the buffer zone around each farm instead of the geographical position to collect the values of environmental variables was used. The determination of the buffer zone size is not a negligible aspect of this analysis since the smaller the area in which parasitological and environmental data are collected, the greater the possibility to make accurate inferences, because averages over large areas can introduce strong ecological bias in correlation studies with infection occurrence data [20]. Sheep and goat flocks sampled in this study are not housed (i.e., grazing all year round), therefore the extraction of predictor values using buffer zones around each farm position resulted the most suitable approach to this case study. However, the reduced grazing area of small ruminants compared to the largest movements of cattle can represent a factor that limits the risk of infection in these farms. Several authors have recently demonstrated that machine learning techniques outperform the traditional statistical techniques such as logistic regression. In fact, this technique has successfully been applied to model the spatial distribution of infectious and parasitic diseases [21,22,23]. A major advantage of RF is that input data do not have to adhere to statistical constraints (e.g., homogeneity of variance, uncorrelated predictor dataset) [24]. In addition, the RF predictive modelling technique leverages the power of the space by the use of distance features which ensure a spatial connotation to the algorithm. A variable representing the distance to hydrographic network could be critical to producing accurate predictions in case studies such as this [18].

The development of forecast models is necessary for a thorough understanding of the spatial components involved in the epidemiology of infection and for a proactive infection management. In particular spatial predictive models, as RF machine learning, are capable of distinguishing areas with high probability of exposure from those with intermediate and low probability [24]. This study is the first attempt to develop a Random Forest predictive model of *C. daubneyi* infection in a Mediterranean area. The use of predictive models moreover allows researchers to take into account not only environmental variables but also climatic factors by considering an extended variety and range of conditions. The future climate change, in fact, if not mitigated, will very likely impact the length of the transmission season and the geographical range of a significant proportion of infectious and parasitic diseases [25].

The free-living stages of *C. daubneyi* and the intermediate molluscan hosts need a temperature range of 10–25 C°, in addition warmer conditions promote the transmission of trematode parasites and raise their local abundance [26] since increasing rainfall and temperatures make the climate more suitable for *G. truncatula* populations [11]. According to Fox et al. [11], the temperature has the major impact in areas where the mean temperature is raised above the 10 C° threshold and rain is not restrictive, instead where temperatures are already above the development threshold of 10 C°, the primary driver becomes changing rainfall patterns. Recently a study conducted on Welsh farms showed sunshine hours as a significant positive predictor for *C. daubneyi*, although the exact reason of this correlation is still unclear [14]. Given the above, climatic variables as mean sunshine hours, mean annual temperature and the number of rainy days need to be investigated, along with environmental variables, for future forecast modelling *C. daubneyi* prevalence at farm level.

Among the variables worth investigating for a more accurate estimate of the spatial distribution of *C. daubneyi*, there are the animal treatments against *F. hepatica*. Jones et al., in a study conducted in UK, showed the regular treatment against *F. hepatica* as significant positive predictor for *C. daubneyi*. In fact, by treating regularly against *F. hepatica*, the number of *F. hepatica* eggs shed onto pasture was reduced, potentially freeing *G. truncatula* snails to be infected with *C. daubneyi* larval stages [14].

The variables positively correlated with the presence of *C. daubneyi* resulted from the ANOVA and Spearman’s tests are highly compatible with the life cycle of the trematoda investigated. Regarding the predictor variables entered into the discriminant model generated by logistic regression, the significance of “average elevation” indicates that areas with medium/high elevation have the higher predicted risk of infection. This result agrees with the significant clusters of infection observed in the western part of the region which is characterized by mountainous/hilly surfaces. The two positive predictors “clayey over fragmental” and “presence of streams and brooks”, as the loamy soil texture resulted from ANOVA and Spearman’s test, could be an indirect measure of the presence of intermediate host snail habitats; the life cycle of *C. daubneyi* in fact involves amphibious snails as intermediate hosts, and thus has strong environmental determinants and strong needs of water. Proximity to water bodies and location in wet soils with poor drainage capacities are considered potential habitats for *G. truncatula* [27]. In particular, the land use (i.e., pasture and wood) and geolithological (impermeable soil) types are indicators of zones where typically there is a presence of water (permanently or temporarily) [8]. In addition, streams and brooks could be also potential movement corridors for *G. truncatula* as reported by Rondelaud et al. [28]. For this reason, environmental information about these gastropod populations is required for knowing in detail the epidemiology of rumen flukes in ruminants.

Furthermore, the extensive farming system of cattle that practice vertical transhumance might influence the parasitic infection transmission patterns. From an epidemiological point of view, this seasonal migration from higher pastures in summer and lower valleys in winter complicates the determination of the areas in which the animals became infected and makes the tracing of the origins and routes followed by the flukes likely impossible [29]. The impact of the animal movements on the spatial distribution of the infection has already been investigated by Ashrafi et al. [30], who established a methodology to assess the altitude influence on *F. hepatica* and *Fasciola gigantica* distribution by grouping the specimens according to altitude ranges. The data on the exact positions where cattle became infected were not available for this study conducted in Basilicata region, but further studies need to be performed to achieve this objective.

This study is part of a surveillance project focused on mapping diseases caused by viral, bacterial and parasitic infections in ruminants in the Basilicata region using GIS. These maps are intended to be used in control programs to prevent and control infections in livestock ruminants in southern Italy.

## 4. Materials and Methods

### 4.1. Study Area and Study Population

This study was carried out from 2016–2017 in the Basilicata region, southern Italy. This region comprises an area of 10,073.32 km^2^ where the provinces of Potenza (40°38′ N; 15°48′ E) and Matera (40°39′ N; 16°36′ E) are located. It is a predominantly mountainous and hilly region; very pronounced altitude differences (from sea level to over 2000 m) and proximity to three different seas (Adriatic to the north-east, Tyrrhenian to the south-west, Ionian to the south-east) attribute a Mediterranean climate to the area. The average temperature in the coldest month (January) is about +8 °C and the warmest month (August) is about +28 °C, with an annual average of +14 °C. Lots of streams and rivers run through the region. The land use is mainly agricultural and pastoral. Livestock farming has a remarkable economic importance in this region. In hilly and mountain areas, for a long time this activity represented a form of use of marginal and uncultivated areas. Currently, 2728 cattle farms (102,984 animals), 5174 sheep farms (183,147.402 animals) and 738 goat farms (1389 animals) [31] still represent a fundamental economic resource for the population of the Basilicata region. In addition to the traditional Podolica cattle breed, which was the only breed bred until the beginning of the last century, others have been added in recent years such as the Marchigiana and Maremmana breeds. The cattle farms are mainly characterized by an extensive farming system with the movement between higher pastures in summer and lower valleys in winter (vertical transhumance).

Sheep and goat farming in the Basilicata region has very ancient origins, representing a role of primary importance in the local economy. Additionally, the small ruminant farms are characterized by an extensive farming system that allows the animals to graze on poor soils with minimal vegetation. Currently, the Merinizzata Italiana, Comisana and Lacoune breeds and mixed-breed are bred in sheep farms, while in goat farms there are only mixed-breed. In recent years, limited grazing areas have been available for small ruminants in this region compared to the past due to the inability of farmers to follow the flocks in larger areas. Agriculture is mostly located in the hills and the most widespread crops are cereals.

### 4.2. Sample Size and Laboratory Procedures

A cross-sectional coprological survey was conducted in 682 sheep, 73 goat and 307 cattle farms distributed in the study area (Figure 5). These farms were randomly selected through veterinarians of the Regional Farmers’ Association of Basilicata [32] on the basis of the availability of the farmers. In each farm, individual faecal samples were collected directly from the rectum of 20 animals according to two age groups: 5 young (0–12 months) and 15 adult (>12 months) animals. The collected samples were stored by vacuum packing [33] and sent refrigerated by courier to the laboratories of the Regional Centre for Monitoring of Parasitosis (CREMOPAR, Campania region, southern Italy). In the laboratory for each farm, four pools of faeces (one for young and three for adults) were prepared, taking 5 g of each individual faecal sample [34,35]. Pooled samples were analyzed by the FLOTAC technique (sensitivity = 94% and specificity = 98%) with a detection limit of 6 EPG using zinc sulphate flotation solution (specific gravity = 1.35) [36,37] (Figure 6).

In addition, in positive faecal samples, a sedimentation technique was used to confirm the diagnosis of *C. daubneyi* and *F. hepatica* based on the color of the eggs [38] (Figure 7). Farms where at least one rumen fluke egg was observed were considered positive.

Parasitological results were used to assess the spatial distribution of infection in cattle, sheep and goat farms, and to develop statistical analysis models to determine environmental risk factors and to delimitate the areas at highest risk of infections in small ruminants (Figure 8).

### 4.3. GIS Construction

Geographical coordinates and *C. daubneyi* infection status (positive/negative) of all surveyed farms were collected, then a shapefile was created using the Arc-GIS Pro 2.7 software platform (ESRI, Redlands, CA, USA). Based on the quartic kernel function, a Kernel density analysis was conducted to estimate the density distribution of rumen fluke infections in cattle and small ruminants. The density of point features around each output raster cell was calculated.

To analyse the possible influence of environmental factors on the presence of infection by *C. daubneyi*, six different variables were evaluated: three orographic variables (altitude, slope and aspect), two variables related to the soil coverage (land use and soil texture) and one hydrographic variable (presence of streams and brooks). The sampled animals spent most of their time on pasture, so a circular buffer zone [7] with a radius of 1 km was generated for each small ruminant farm. It was not possible to establish an accurate buffer zone size for cattle farms since grazing seasonality (vertical transhumance) and distances differ according to the farming system. For this reason, the statistical and predictive analyses were conducted only for small ruminant farms in this study. Slope and aspect maps were derived from the digital terrain model (elevation data). These data-layers were reclassified to simplify the information in their raster, by grouping values. The elevation was divided in low (0–500 m), medium (500–1000 m), high (1000–1500 m) and very high (>1500 m). The slope was divided in flat (0–5°), low (5–15°), medium (15–30°) and high (30–60°). Finally, the aspect was divided in north (337.5–360° and 0–22.5°), north-east (22.5–67.5°), east (67.5–112.5°), south-east (112.5–157.5°), south (157.5–202.5°), south-west (202.5–247.5°), west (247.5–292.5°) and north-west (292.5–337.5°). For each class of these three data layers, the number of cells within the buffer zones, the average and the standard deviation were calculated. Subsequently, the number of cells of 34 land cover classes within the 755 buffer zones was counted. The same operation was executed for the cells of 13 soil texture classes and for hydrographic network data layer. All spatial data-layers were projected to WGS84_UTM_33N and converted to raster file with a cell size of 100 m (Figure 9). The environmental variables considered are listed in Table 4.

### 4.4. Predicted Distribution of C. daubneyi

The Getis-Ord-Gi* statistic [39,40] was used to develop a Hot Spot analysis, in order to determine whether the observed spatial clustering of rumen fluke-positive farms was more pronounced than expected in a random distribution scenario. The standard deviation represented by the Z score was a measure used to represent the intensity of the clustering: a higher value of the Z score indicated that, rather than a random pattern, the farms exhibited statistically significant clustering. Different confidence levels (%) and significance levels (*p*-value) applied to specific standard deviation values: to z scores <−1.65 or >+1.65 corresponded a 90% upper confidence level (CL_U_) and a *p*-value <0.10, to z-scores <−1.96 or >+1.96 corresponded a 95% CL_U_ and a *p*-value <0.05 and to z-scores <−2.58 or >+2.58 corresponded a 99% CL_U_ and a *p*-value <0.01 [41].

The prediction of presence/absence of *C. daubneyi* in the study area was performed by the development of a supervised machine learning technique of Random Forest (RF). The model was constructed by using the Forest-based classification and regression tool (ArcGIS Pro 2.7), which is an adaptation of Leo Breiman’s Random Forest algorithm [42]. Firstly, points data of presence/absence of *C. daubneyi* were used to train the model, except for a random subset of input data (10%) which was excluded from the training and used for the validation. The environmental variables included in GIS were used as categorical explanatory training variables. Only the hydrographic network was not used as a categorical training variable but as explanatory training distance feature; since the ArcGIS tool allowed us to calculate the distance from natural watercourses to the farms. The decision trees were generated using randomly selected data from the input dataset and each tree established a relationship between explanatory training variables and the categorical variable to predict. In the second prediction step, all the outcomes generated from each decision tree were used by the model to predict the risk of the infection on the study area and to predict unknown values at other locations characterized by the same environmental explanatory variables. The importance of the variables was calculated using Gini coefficients, which can be thought of as the number of times a variable is responsible for a split and the impact of that split divided by the number of trees. Splits are each individual decision within a decision tree [18].

### 4.5. Statistical Analysis

The univariate statistical analysis of Spearman’s (*r_s_*) was conducted to assess the strength of association between the independent variables and the dependent variable represented by the infection status (positive or negative to *C. daubneyi*) of each farm examined. The Spearman’s (*r_s_*) was performed since the environmental variables were not normally distributed. In addition, a one-way ANOVA univariate analysis was conducted to compare the average values of the variables and consequently to determine if there was a statistically significant difference between positive and negative buffer zones. The variables that showed significance in both Spearman’s (*r_s_*) and the one-way ANOVA tests were used as predictors in a stepwise logistic regression analysis; the forward method used to select the variables, consisted of an entry test based on the significance of the score statistic and a removal test based on the probability of the Wald statistic.

## 5. Conclusions

The findings discussed above suggest that a combined use of different geostatistical techniques improve the *C. daubneyi* infection risk prediction in small ruminants.

This study also showed the efficacy of GIS for monitoring the spatial distribution of rumen flukes in livestock ruminants, providing useful tools to predict the parasitic infection in the study area and at other locations characterized by the same environmental explanatory variables.

## Figures and Tables

**Figure 1 pathogens-10-01122-f001:**
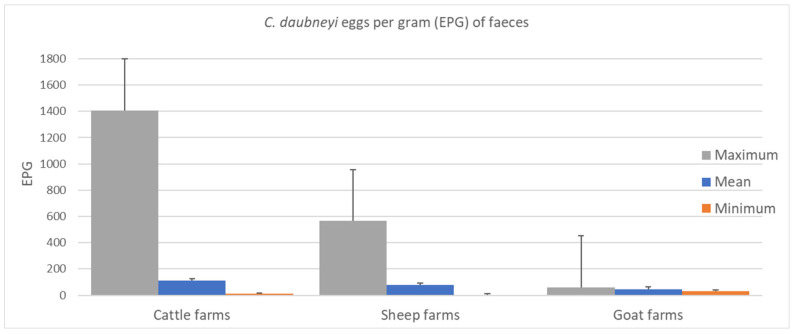
Variability of *C. daubneyi* egg counts in cattle, sheep and goat farms: mean (blue), maximum (grey) and minimum (orange) eggs per gram (EPG) of faeces and standard errors.

**Figure 2 pathogens-10-01122-f002:**
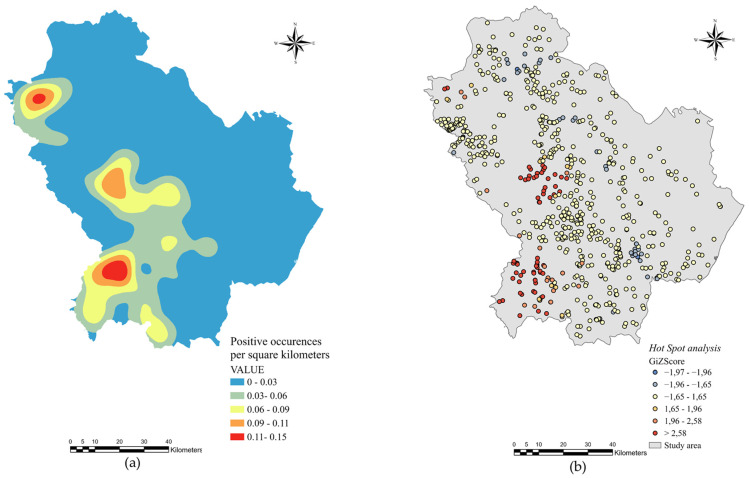
Spatial analysis of *C. daubneyi* distribution in the study area: (**a**) Kernel density analysis of sheep, goat and cattle farms examined; (**b**) Hot Spot analysis for clusters of small ruminant farms positive to rumen flukes.

**Figure 3 pathogens-10-01122-f003:**
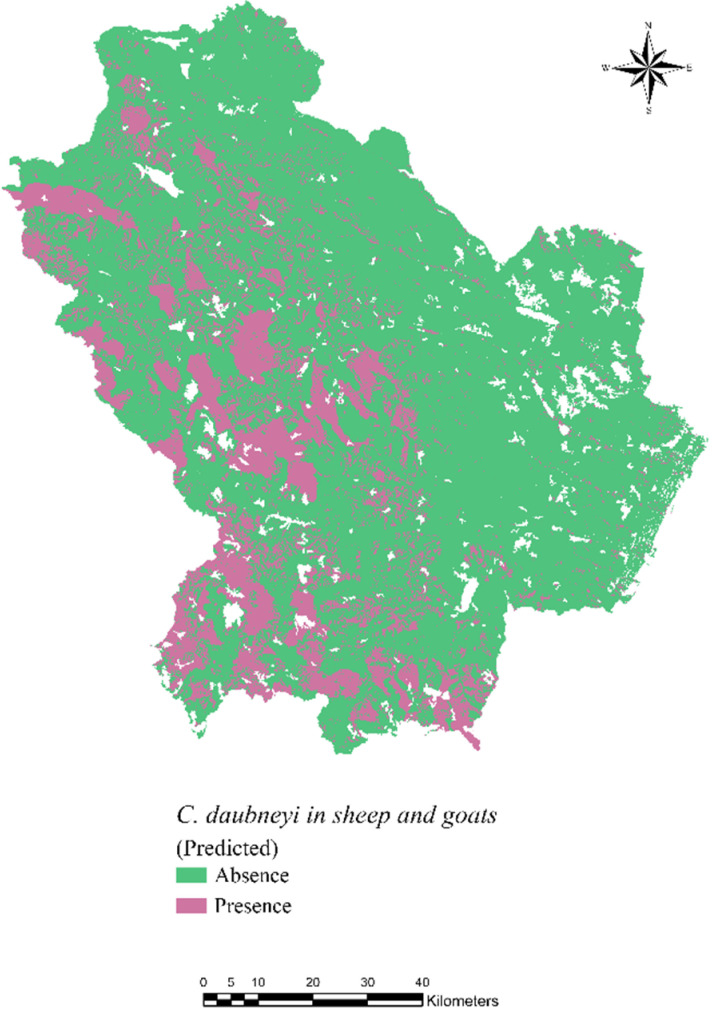
The predicted spatial distribution of *C. daubneyi* generated by the supervised Forest-based machine learning algorithm.

**Figure 4 pathogens-10-01122-f004:**
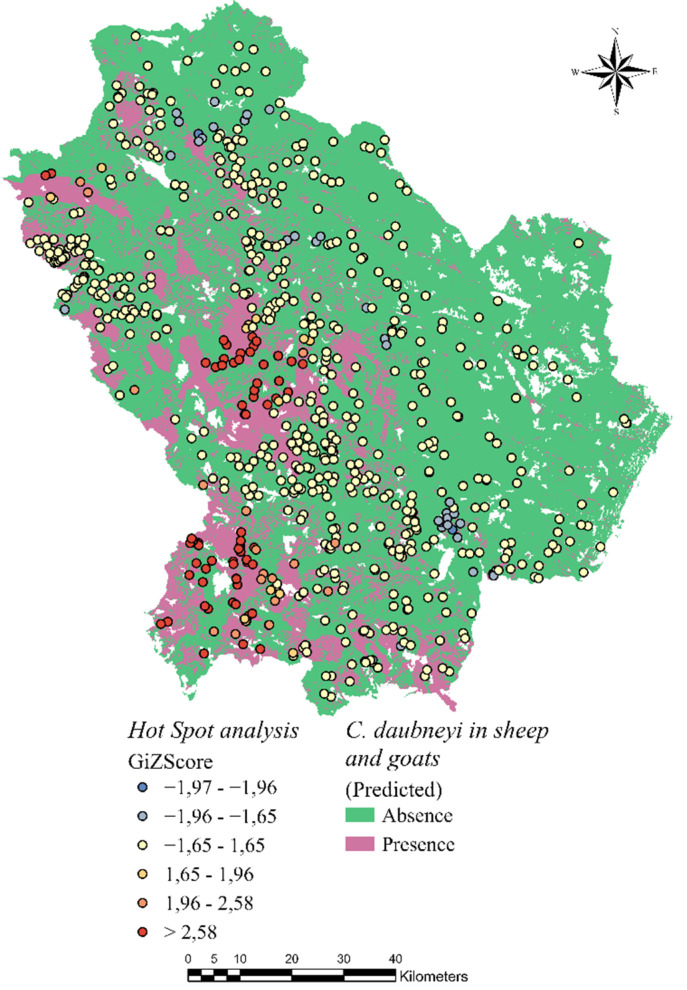
The results for clustering obtained from the Hot Spot analysis and the RF algorithm.

**Figure 5 pathogens-10-01122-f005:**
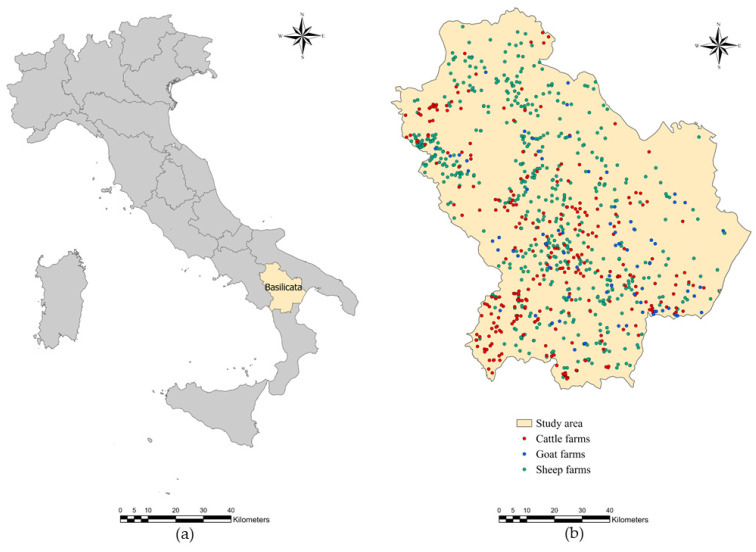
Study area and data: (**a**) Administrative regional boundaries of Italy; (**b**) Location of sampled farms in the Basilicata region.

**Figure 6 pathogens-10-01122-f006:**
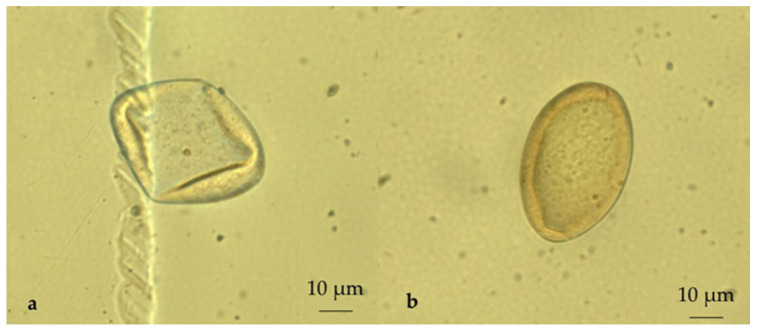
Comparison of *Calicophoron daubneyi* (**a**) and *Fasciola hepatica* (**b**) eggshells (broken eggs due to flotation solution) under FLOTAC (400× magnification): *F. hepatica* eggshells are yellow-brown, whereas *C. daubneyi* eggshells are clear.

**Figure 7 pathogens-10-01122-f007:**
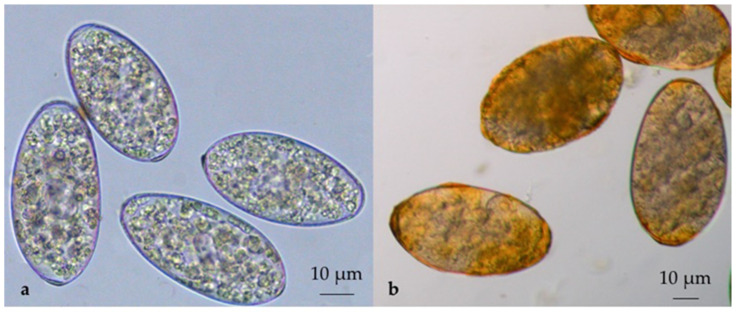
Comparison of *Calicophoron daubneyi* (**a**) and *Fasciola hepatica* (**b**) eggs in aqueous suspension (400× magnification): *F. hepatica* eggs have the yellow-brown shells, whereas *C. daubneyi* eggs have clear shells.

**Figure 8 pathogens-10-01122-f008:**
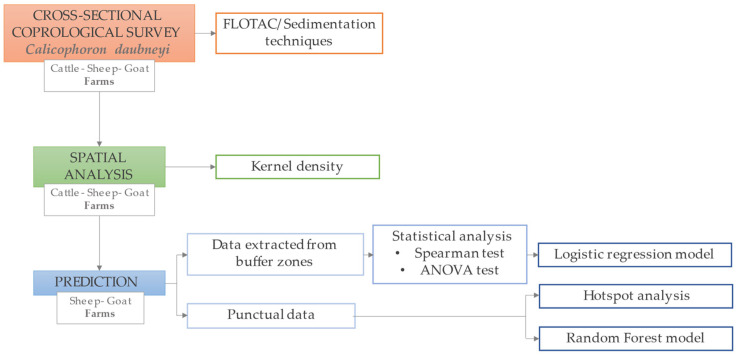
Study design: cross-sectional coprological survey conducted in sheep, goat and cattle farms to estimate the prevalence of *Calicophoron daubneyi* and to evaluate its spatial distribution in the Basilicata region of southern Italy; statistical and predictive tests were performed in order to determine environmental risk factors of infection only for sheep and goat farms.

**Figure 9 pathogens-10-01122-f009:**
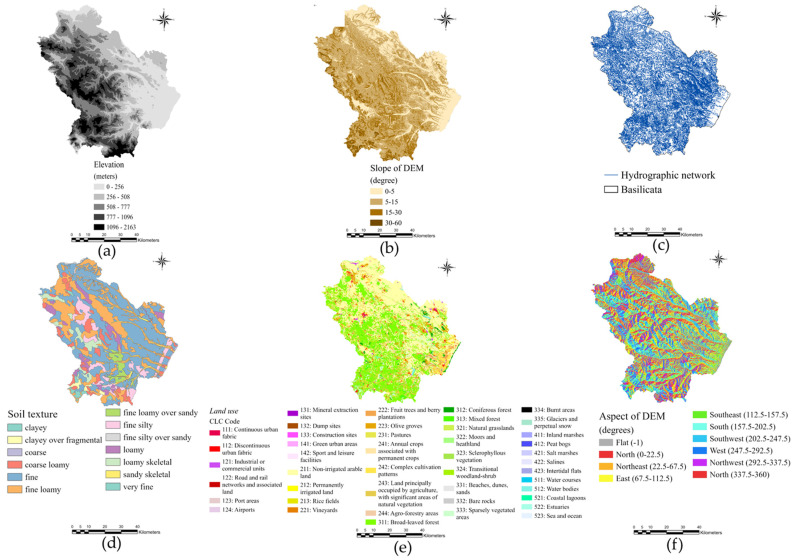
Environmental data-layers selected for GIS model construction: (**a**) Elevation; (**b)** Slope; (**c**) Hydrographic network; (**d**) Soil texture; (**e**) Land use; (**f**) Aspect.

**Table 1 pathogens-10-01122-t001:** Top variable importance table. The values in the Importance column are the sum of the Gini coefficients from all the trees for each variable listed. The values in the % column are the percentage of the total sum of Gini coefficients.

Variable	Importance	%
Soil texture	1.52	21
Land use	1.45	20
Hydrographic network	1.42	19
Aspect	1.41	19
Slope	0.79	11
Elevation	0.79	11

**Table 2 pathogens-10-01122-t002:** Results of the Spearman’s correlation analysis and one-way ANOVA test. The values in the column *r_s_* indicate the correlation between the positive farms and the environmental variables. In the column *F* value are the results of the *F*-statistic test. The statistical significance of results is shown by the values in the column *p* value.

Variable	Spearman’s Test		One-Way ANOVA Test	
	*r_s_*	*p* Value	*F* Value	*p* Value
Pastures	0.158	<0.001	6.104	0.014
Broad-leaved forest	0.128	<0.001	7.528	0.060
Transitional woodland-shrub	0.096	<0.001	5.739	0.017
Medium elevation (500–100 masl)	0.074	<0.05	7.429	0.008
High elevation	0.132	<0.001	4.683	0.034
Average elevation	0.151	<0.001	19.654	<0.001
Standard deviation of elevation	0.097	<0.001	6.315	0.012
Low slope (5°–15°)	0.087	<0.001	13.373	<0.001
Average slope	0.083	<0.05	17.680	<0.001
Coarse loamy	0.114	<0.001	4.580	0.036
Loamy skeletal	0.137	<0.001	4.788	0.033
Clayey over fragmental	0.081	<0.05	8.597	0.003
Presence of streams and brooks	0.102	<0.001	8.679	0.003

**Table 3 pathogens-10-01122-t003:** Results of stepwise logistic regression.

Variable	*p*	Exp (B)	95% CI
Average elevation	0.003	2.78	1.58–4.88
Clayey over fragmental	<0.001	1.01	1.00–1.01
Presence of streams and brooks	0.009	1.01	1.00–1.02

* *p* < 0.05.

**Table 4 pathogens-10-01122-t004:** Sources and data-layers included in geographical information system for modelling the spatial distribution of *Calicophoron daubneyi* in the Basilicata region of southern Italy.

Name	Type and Resolution	Year	Source	Description
The Corine Land Cover	Vector layer (100 m)	2018	Copernicus	Inventory on land cover of EU (44 classes)
Regional boundaries	Vector data	2011	ISAT	The regional administrative boundaries
Hydrographic network	Vector data (1:5000)	2015	Geodatabase of region Basilicata	Representation of natural watercourses
Elevation (DTM)	Raster (100 m)	2016	Geodatabase of region Basilicata	Digital elevation model of the region
Slope	Raster (100 m)	2016	Geodatabase of region Basilicata	The rate of change of elevation for each cell
Aspect	Raster (100 m)	2016	Geodatabase of region Basilicata	The direction of the compass facing a slope
Soil texture	Vector (100 m)	2012	Geodatabase of region Basilicata	Distribution of the mineral particles of the soil, according to the Soil Taxonomy

## Data Availability

The raw data supporting the conclusions of this article will be made available by the authors, without undue reservation.

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
