# Peer review of "Mapping the Spatial Distribution of the Rumen Fluke Calicophoron daubneyi in a Mediterranean Area"

_pathogens, 2021, doi:10.3390/pathogens10091122_

Round 1

Reviewer 1 Report

The paper entitled Mapping the spatial distribution of the rumen fluke (Calicophoron daubneyi) in a Mediterranean area contributes enormously to the knowledge of this emerging parasite and is the work of a respected research group in the area.

However of the importance of the work, the authors are unsuccessful in explaining the significance of such, and omit relevant facts that explain the relevance of the work and also would deliver a better read.

The paper gives an insight into three ruminant species, a proper description of the herds/flocks by species would be appropriate, this is particularly important as local breeds inhabit the area and scarce information of such breeds is currently available.

Introduction

In the introduction of the text the importance of studying the spatial distribution of rumen fluke is presented in a superficial way, for example, how does the disease affect animals? Is it sub-acute? Is there death related to it? Economic importance, if known?  how would this affect the local? - The importance of the emerging disease is not clear

Line 37: which climatic changes could be contributing to the emergence.

Line 38: write ‘to be completed’ after ‘(Galba truncatula)’.

Line 50: how does this monitoring activities work.

Line 52-55: Re write aims appropriately.

Results

This section of the paper does not explain in the text the totality of results, at least the most important findings of each analysis must be added to the text to complement the tables and figures. Some of the results have been added in the discussion, this is not appropriate and it should be amended,

59: ‘faeces’ instead of ‘feces’

59-61: This information is very interesting. A bar graph with intervals would make it easier to understand.

65: Add ‘of goat and sheep farms’ after ‘The hotspot analysis’

69-72: Please, briefly explain results from the RF algorithm analysis

77-80: Briefly explain results from Spearman’s rho, one way ANOVA and stepwise logistic regression

Discussion

111-114: Any information on goats infection?

116-117: where in Italy? These references are talking about 10% difference to the finding in the present work, how are these similar?

118-121: farm management, timing, and type of flukicide treatment, size of grazing areas, housing periods, production system, breed, animal density and age group affect the prevalence and fluke burden of C. daubneyi in cattle. The relevance of these factors to the current work are significant, but no explanation of how they are related is provided by the authors, please amend. Also, the authors could explain local factors that might be also affecting and compare them to relevant literature.

122: how are those results inconsistent?

125-126: Authors indicate that hotspot mapping ‘does not provide information about factors that are considered to cause the phenomenon’. This raises doubts on the relevance of this section of the results.

177-179: This text corresponds to results. Please discuss this result in this section.

181: The paragraph starting in this line corresponds to results, results are being presented in the discussion section of the paper. Discuss this results in this section. Please review and correct.

189: The biological characteristics of the parasite should be explained in the introduction to give a better understanding to the reader of the importance of environmental factors for the disease.

Material and methods

 Line 205: ad to the subtitle ‘and study population'

In this part of the text a description of the study population must be included for the ruminant species studied, include breeds, distribution, etc. Specially considering that the statistical and predictive analyses were carried out only small ruminants.

Line 221: Any information on the economic importance of livestock farming in the region?

Line 229: why are the grazing areas for small ruminant diminishing in the area?

Lines 234-234: how were the farms recruited?

Lines 237: How were the samples sent and conserved?

Lines 239-244: Add sensitivity and specificity of the tests

Line 243: How were the eggs differentiated from F.hepatica and other rumen fluke species?

267: Different cattle breed graze different size pastures according to breed? Please explain or rewrite.

267: ‘were’ instead of ‘was’

308-310: Please add some commas, rewrite.

326: How was the distribution of all data determined?

331: What was the stepwise logistic regression used for?

Conclusion

334: Please remove ‘can’

Figure 5: indicates that cattle, sheep and goat data were used in the three big areas of the study. However, authors indicate on lines 267-268 that only small ruminants data was used because of buffer zones, please correct and revise the complete text to make clear in which instances cattle, sheep and goat data was used and where sheep and goat data was used.

Reviewer 2 Report

This is an interesting and useful study which should be published. There are, however, a few aspects which should be improved and several considerations which should be added.

1.- The Introduction written in a unique paragraph is hard to read. It should be broken into at least three paragraphs.

2.- Figure 4(a) should be moved to Figure 1 to allow the reader understand which is the Italian region in question in Figures 1 and the subsequent Figures 2 and 3. This is of course a consequence of having the M&M section at the end in this journal.

3.- Line 216: Entry of the line at left should be arranged.

4.- Lines 242-243: It is noted that "a sedimentation technique was 242 used to differentiate the C. daubneyi eggs from those of F. hepatica". Difficult to understand that eggs from these two species can be distinguished by a sedimentation technique. Is it there a difference in the sedimentation speed? Or perhaps the eggs of C. daubneyi remain floating whereas those of Fasciola fall down? Authors should concretely clarify how they differentiated the eggs from one species from those of the other. Adding a figure including an egg of each species would facilitate the understanding.

5.- English writing should be uniformized whether to British or American.

6.- Line 172: Increasing temperatures may or may not favour G. truncatula populations. In northern Europe an increase of temperatures may be favourable, but at the Mediterranean shore they may act against. The minimum and maximum temperature thresholds of G. truncatula should be considered here.

7.- Nothing is mentioned about animal treatments against Fasciola in the study area. May fasciolicides have an action against C. daubneyi? Something should be added concerning this.

8.- Line 226: Vertical transhumance is noted to be followed in this region, i.e. movements between higher pastures in summer and lower valleys in winter. This poses the main question to this interesting study: the animals become infected in the highlands, in the lowlands or in both places? It may happen that they become infected in the highlands and afterwards show their infection in the lowlands. Or viceversa. If one of these is the case, this would mean a bias in the results, i.e. the trematode being detected but in fact not present in the place studied (i.e., its life cycle not occurring there). This phenomenon due to vertical transhumance has been verified to occur in Fasciola. See the following articles:

Mas-Coma et al., 2009. Fasciola, lymnaeids and human fascioliasis, with a global overview on disease transmission, epidemiology, evolutionary genetics, molecular epidemiology and control. Advances in Parasitology, 69: 41-146.

Ashrafi et al.,  2015. Distribution of Fasciola hepatica and F. gigantica in the endemic area of Guilan, Iran: relationships between zonal overlap and phenotypic traits. Infection, Genetics and Evolution, 31: 95-109.

9.- Regarding the aforementioned question, it would be important to consider the life span of the adult stage of C. daubneyi inside each one of the livestock species studied, if known. For instance, F. hepatica lives up to 13 years in sheep, which means many years of vertical transhumance.

10.- Considering the existence of vertical transhumance, the model would be nice to be build on data from monthly surveys. We know today that trematodes shed more eggs during the initial months, both in primoinfection and in reinfections, and this could be detected in monthly surveys. See following article:

Valero et al., 2020. Impact of fascioliasis reinfection on Fasciola hepatica egg shedding: relationship with the immune-regulatory response. Acta Tropica, 209: 105518.
